# An empirical evaluation of Lex/Yacc and ANTLR parser generation tools

**Francisco Ortin** [1,2]*, **Jose Quiroga** [1], **Oscar Rodriguez-Prieto** [1], **Miguel Garcia** [1]

**1** University of Oviedo, Computer Science Department, Oviedo, Spain, **2** Munster Technological University, Cork Institute of Technology, Computer Science Department, Rossa Avenue, Bishopstown, Cork, Ireland

* ortin@uniovi.es, francisco.ortin@cit.ie

## Abstract

Parsers are used in different software development scenarios such as compiler construction, data format processing, machine-level translation, and natural language processing. Due to the widespread usage of parsers, there exist different tools aimed at automizing their generation. Two of the most common parser generation tools are the classic Lex/Yacc and ANTLR. Even though ANTLR provides more advanced features, Lex/Yacc is still the preferred choice in many university courses. There exist different qualitative comparisons of the features provided by both approaches, but no study evaluates empirical features such as language implementor productivity and tool simplicity, intuitiveness, and maintainability. In this article, we present such an empirical study by conducting an experiment with undergraduate students of a Software Engineering degree. Two random groups of students implement the same language using a different parser generator, and we statistically compare their performance with different measures. Under the context of the academic study conducted, ANTLR has shown significant differences for most of the empirical features measured.

**Data Availability Statement:** All the raw data obtained from conducting all the experiments are available for download from https://reflection.uniovi.es/download/2022/plos-one.

## Introduction

Parsing—also known as syntax or syntactic analysis—is the process of analyzing a string of terminal symbols conforming to the rules of a formal grammar [1]. Such a grammar may describe a natural language (*e.g.*, English or French), a computer programming language, or even a data format. The process of recognizing the terminal symbols, also called tokens, from a sequence of characters is called lexical analysis [2]. Lexical analyzers are commonly called lexers or scanners.

Parsers are software components that, using a lexer to recognize the terminal symbols of a given language, analyze an input, check its correct syntax, and build a tree that represents the input program with a hierarchical data structure [1]. Parsers are used for different tasks in computer science such as performing machine-level translation of programs (*e.g.*, (de)compilers, transpilers, and (dis)assemblers), creating software development tools (*e.g.*, profilers, debuggers, and linkers), natural language processing (*e.g.*, dependency and constituency parsing), and data format processing (*e.g.*, JSON, GraphViz DOT, and DNS zone files).

**Funding:** This work has been partially funded by the Spanish Department of Science, Innovation, and Universities: project RTI2018-099235-B-I00. The authors have also received funds from the University of Oviedo through its support to official research groups (GR-2011-0040).

**Competing interests:** The authors have declared that no competing interests exist.

The importance of parsing in computer science has implied the existence of different tools aimed at the automatic generation of parsers [3]. By using these tools, the user describes the syntax of the language to be recognized, and the tool generates a parser for the specified language. There exist plenty of parser generators, such as Yacc/Bison, ANTLR, JavaCC, LISA, SableCC, and Coco/R, among many others [4]—some of them provide additional features for compiler construction beyond parsing.

Although these tools have different properties such as the parsing strategy or the programming language of the generated code, most software developers opt for either the classic Lex/Yacc approach or the ANTLR 4 parser generator [5]. Parser generators are also used to teach programming language and compiler implementation courses in computer science degrees. Table 1 shows the parser generation tools used in the compiler construction courses of the top 10 universities, according to the four following rankings (computer science sections): Shanghai Consultancy [6], Quacquarelli Symonds [7], Times Higher Education [8], and US News [9].

As shown in Table 1, most courses (13 out of 16) use variants of the classic Lex/Yacc approach: Bison is the GNU implementation of Yacc; Flex is a free and open-source version of Lex; OCamlLex, OCamlYacc, and Menhir are Lex/Yacc implementations for the OCaml programming language; ML-Lex and ML-Yacc for standard ML; and JFlex and CUP for Java. Only the Computer Language Engineering course delivered at MIT uses ANTLR, despite its advanced features (see next sections) and its widespread usage in the development of different products such as Hibernate, Twitter search, Hadoop, and Oracle SQL developer IDE [10].

Due to the existing differences between Lex/Yacc and ANTLR, the main contribution of this article is an empirical comparison of these two parser generation tools. To that aim, we randomly divide into two groups the students of a Programming Language Design and Implementation course in a Software Engineering degree. Each group is taught a different parser generator and requested to develop an imperative procedural programming language. We then measure different variables considering the students' performance when using each tool,

**Table 1. Parser generation tools used in the compiler courses of the top 10 universities according to Shanghai Consultancy [6], Quacquarelli Symonds [7], Times Higher Education [8], and US News [9] rankings.**

| University | Course name | Parser generator used |
|---|---|---|
| Carnegie Mellon University | Compiler design | ML-Lex/ML-Yacc |
| ETH Zurich | Compiler design | OCamlLex/Menhir |
| Harvard University | Compilers | OCamlLex/Menhir |
| Imperial College London | Compilers | Lex/Yacc |
| Massachusetts Institute of Technology | Computer language engineering | ANTLR |
| Nanyang Technological University | Compiler technology of programming languages | Lex/Yacc |
| National University of Singapore | Compiler design | JFlex/CUP |
| Peking University | Principles of Programming Language Design | OCamlLex/Menhir |
| Stanford University | Compilers | Flex/Bison |
| Swiss Federal Institute of Technology in Lausanne | Computer language processing | Manual implementation with a LL(1) grammar |
| The Chinese University of Hong Kong | Compiler Construction | JavaCC |
| Tsinghua University | Principles of assembly and compilation | Lex/Yacc |
| University of California, Berkeley | Programming languages and compilers | JFlex/CUP |
| University of Cambridge | Compiler construction | OCamlLex/OCamlYacc |
| University of Oxford | Compilers | OCamlLex/OCamlYacc |
| University of Toronto | Compiler and interpreters | Flex/Bison |

Only 16 universities appear in Table 1 because some of them are within the top 10 positions in more than one ranking. If information about the parser generator is not provided, the university is not included. Universities are ordered alphabetically.

and their opinion about the parser generator used. These data provide evidence, under the context of our study, to undertake an empirical comparison of the two parser generators.

The following section discusses the related work. Then, we present a qualitative comparison of Lex/Yacc and ANTLR. For the quantitative comparison, we describe the methods used in our study, the results, and different discussions. Finally, we present the conclusions of our work.

## Related work

Daniela da Cruz *et al*. present a qualitative comparison of Yacc, LISA, and ANTLR 3 parser generators [11]. To that aim, they define the Lavanda Domain-Specific Language (DSL) whose goal is to compute the number of laundry bags a company daily needs to send to wash. They implement Lavanda as an interpreter with each of the three tools. The Lavanda semantics is specified as an attribute grammar that is translated to Yacc, LISA, and ANTLR 3. Based on the three implementations, the authors present a qualitative evaluation where they analyze different features divided into three groups: language specification, parser generator, and generated code. Overall, they state that Yacc is the poorest tool, and ANTLR 3 and LISA are very similar, regarding the comparison criteria utilized in their study.

Klint *et al*. perform an empirical analysis of the benefits of using tools for the implementation of DSLs [12]. In their study, they select six experts to implement WAEBRIC, a little language for XHTML markup generation used for website creation. Each expert develops a different language implementation: Java, C#, and JavaScript implementation with no tools; and the other three using ANTLR, Microsoft "M", and OMeta. For the comparison, they measure different quantitative variables such as volume (number of modules, units, and lines of code), cyclomatic complexity, and percentage of code duplication. They also present a qualitative evaluation comparing the maintainability of each implementation. Their evaluation concludes that the maintainability of the DSL implementations is higher when generators are used.

Language workbenches are tools that lower the development costs of implementing new languages and their associated tools [13]. They include different components, such as parser generators, to make language-oriented programming environments practical [14]. Erdweg *et al*. present empirical data on the implementation of a tiny Questionnaire Language (QL) to create questionnaires forms using ten language workbenches (Ensō, Más, MetaEdit+, MPS, Onion, Rascal, Spoofax, SugarJ, Whole, and Xtext) [15]. First, they analyze to what extent each language workbench is able to support all the features of the QL language. Then, they measure the source lines of code and the number of model elements used. No language workbench realizes all the features. The features less widely supported are type checking, DSL program debugging, quick fixes, and live translation.

Josef Grosch performs a qualitative and quantitative comparison of lexer (Lex, Flex, and Rex) and parser generators (Yacc, Bison, PGS, Lalr, and Ell) [16]. He first presents a qualitative comparison of both kinds of tools by analyzing their features. Then, Grosch compares the size of the lexer/parser table and the generated program, generation time, and execution time of the code produced. For lexical analysis, Rex is the tool with the lowest generation times and table and lexer sizes, and it generates the fastest lexers. For the parser generation tools, Bison provides the lowest generation times, and table and parser sizes, whereas Ell generates the fastest parsers.

Hyacc is a Yacc-compatible open-source parser generator that accepts all LR(1) grammars and generates full LR(0), LALR(1), LR(1), and partial LR(k) parsers [17]. Using Hyacc, Chen and Pager perform an empirical comparison of runtime performance and memory consumption of the most common LR(1) parsing algorithms [18]. For the evaluation, they used 17

simple grammars and 13 specifications of real programming languages. The Knuth LR(1) algorithm uses more memory than the rest of the algorithms, and Bison LALR(1) is the one with the lowest memory consumption. Regarding runtime performance, Knuth LR(1) is the slowest algorithm and LR(0) runs the fastest.

Renée McCauley performs a shallow analysis of parser generation tools and how they could be used in the delivery of different courses beyond programming language design and implementation, such as operating systems, software engineering, and even human-computer interaction [19]. Parser generation tools represent a good example of principles that have endured, because the underlying theory and algorithms are well understood and used to implement different kinds of tools. McCauley indicates that ANTLR is more powerful than the traditional Lex/Yacc approach, mainly because of its lexer and parser integration, automatic parse tree generation, and its tree walker grammars.

## Features of Lex/Yacc and ANTLR

Table 2 compares the main features of the two parser generators. Yacc follows a bottom-up parsing strategy, creating a node of the parse tree only when its child nodes have been created [2]. It implements an LALR(1) (Look-Ahead LR) parser, which is a simplification of general LR(1) bottom-up parsers [20]. To know whether the parser must analyze another token from the input or reduce one production from the grammar, the algorithm analyzes one single token (lookahead).

On the contrary, ANTLR follows a top-down parsing strategy, creating the parent nodes of the parse tree before their children. It implements an adaptive LL(*) parsing algorithm [21]

**Table 2. Qualitative differences between Lex/Yacc and ANTLR.**

| Feature | Lex/Yac | ANTLR 4 |
|---|---|---|
| Parsing strategy | Bottom-up | Top-down |
| Parsing algorithm | LALR(1)/LR(1) (Menhir) | Adaptive LL(*) |
| Lookahead tokens | 1 | Adaptive finite (*) |
| Theoretical computational complexity | $O(n)$ | $O(n^4)$ |
| Grammar notation | BNF | EBNF |
| Automatic tree generation | No | Yes |
| Lexer algorithm | DFA | Adaptive LL(*) |
| Joint lexer and parser specification | No | Yes |
| Output languages | C (Lex/Yacc, Flex/Bison), Java (JFlex/CUP), OCaml (OcamlLex/OcamlYacc, Menhir), ML (ML-Lex/ML-Yacc) | Java, C#, C++, Python2, JavaScript, Python3, Swift, Go |
| Semantic actions | Embedded | Embedded and separated |
| Right recursion | Yes | Yes |
| Left recursion | Yes | Only direct |
| Semantic predicates | No | Yes |
| Plugin support | No | IntelliJ, NetBeans, Eclipse, Visual Studio |
| Testing tool | No | Yes (TestRig) |
| License | Proprietary (Lex), CPL (Yacc), GNU GPL (Bison), QPL (OCamlLex, OCamlYacc, Menhir), BSD (Flex, CUP, JFlex) | BSD |

that analyzes a finite but not fixed number of tokens (lookahead) [22]. At parsing, it launches different subparsers when multiple productions could be derived, exploring all possible paths. Then, the subparser with the minimum lookahead depth that uniquely predicts a production is selected [22]. This algorithm has an $O(n^4)$ theoretical computation complexity—$n$ being the number of nodes in the parse tree—, but it has been experimentally measured to be linear in both computation and space for real programming language implementations [22].

Yacc uses the BNF (Backus-Naur Form) notation to specify language grammars [23], while ANTLR supports an extended version that includes different operators including those provided in common regular expressions [10]. This feature allows language implementors to describe grammars with fewer productions. Moreover, simpler grammars produce smaller parse trees, which are automatically generated by ANTLR (Yacc does not support this feature). ANTLR uses the same grammar notation and parsing strategy for both lexical and syntactic language specifications. Yacc only provides syntax specification, while lexical analysis should be implemented apart, commonly using Lex, which generates Deterministic Finite Automata (DFA) [23].

One language specification in ANTLR can be used to generate parsers in different programming languages (see Table 2). It provides listeners to decouple grammars from application code (semantic actions), encapsulating its behavior in one module apart from the grammar, instead of fracturing and dispersing it across the grammar. Unfortunately, Yacc does not provide this option, so the user must write the semantic actions embedded across the grammar.

Both tools provide right recursion. ANTLR 4 has improved one of its main limitations of the previous versions: the support of direct left recursion. Top-down parsers do not provide left recursion, but ANTLR 4 rewrites direct left recursive productions into non-left-recursive equivalents to support direct left recursion [10]. This is an important benefit to allow writing the classic expression productions with left and right direct recursion.

ANTLR provides semantic predicates to resolve some grammar ambiguities. For example, `f(i)` in Visual Basic could mean array indexing and function invocation. By storing the type of `f` upon its definition, a semantic predicate could select the appropriate production by consulting the symbol table. As shown in Table 2, ANTLR has different plugins for existing IDEs to help the user in the language implementation process. It also provides TestRig, a testing tool that displays information about how the parser and lexer work at runtime, and different views of the parse tree for a given input.

## Methods

After a brief qualitative comparison of Lex/Yacc and ANTLR, we describe the methodology we used for the empirical comparison. We followed the guidelines by Kitchenham *et al*. for empirical research works in software engineering [24].

### Context

We conducted an experiment with undergraduate students of a Programming Language Design and Implementation course [25] in a Software Engineering degree [26], at the University of Oviedo (Spain). The course is delivered in the second semester of the third year. The students have previously attended five programming courses, three of them in Java, and the other two in C# and Python. Such courses cover different topics such as procedural, object-oriented, functional, and concurrent programming [27], and human-computer interaction [26]. They have also taken other courses that include Java programming assignments such as software design, algorithmics, data structures, web development, databases, numeric computation, distributed systems, information repositories, computer architecture, and operating systems

[26]. In four of these courses, the students have to implement a real Java application following a project-based learning approach.

The Programming Language Design and Implementation course is delivered through lectures and laboratory classes, summing 58 class hours (30 hours for labs and 28 for lectures) along the second semester (6 ECTS). Lectures introduce the theoretical concepts of programming language design. Labs (Table 3) follow a project-based learning approach, and they are mainly aimed at implementing a compiler of a medium-complexity imperative programming language. That language provides integer, real and character built-in types, arrays, structs, functions, global and local variables, arithmetical, logical and comparison expressions, literals, identifiers, and conditional and iterative control flow statements [25]. The compiler is implemented in the Java programming language.

The course has two parts. In the first seven weeks, the students have to implement the lexical and syntactic analyzers, using a generation tool. The parser should return an Abstract Syntax Tree (AST) if no errors are detected for the input program. In lab 7, it is checked that the work of students is completed, giving detailed feedback about the mistakes they need to correct.

In the second part of the course, they traverse the AST to perform semantic analysis and code generation for a stack-based virtual machine [30]. The evaluation consists of a theory exam (50%) and a lab exam where the students must extend their language implementation (50%). Both evaluations are performed after delivering all the lectures and labs.

Our study took place in the second semester of the 2020-2021 academic year. The 183 students (131 males and 52 females) enrolled in the course were divided into two random groups: the first group had 92 students (67 males, 25 females, average age 22.8, and standard deviation 2.04) and the second one was 91 (64 males, 27 females, average age 22.7, and standard deviation 2.13). Retakers were not included in our study, because they have previous knowledge about the course and experience in the utilization of Lex/Yacc (the tool used in the previous years).

For the first part of the course (the first seven weeks), lectures and labs for the first group were delivered using BYaccJ/JFlex (Java versions of the classic Yacc/Lex generators). For the second group, ANTLR was used instead. The second part of the course has the same contents for both groups, since there is no dependency on any generation tool. Both groups implemented the very same programming language, and they had the same theory and lab exams.

**Table 3. Lab distribution per weeks.**

|  | Week | Topic |
|---|---|---|
| First part | 1 | Architecture of a language implementation. |
|  | 2-3 | Abstract Syntax Tree (AST) design. |
|  | 4 | Implementation of a lexical analyzer with JFlex or ANTLR. |
|  | 5 | Implementation of a syntax analyzer with BYaccJ or ANTLR. |
|  | 6 | Creation of an AST instance for the input program while parsing, using BYaccJ or ANTLR. |
|  | 7 | First part completion. It is checked that the student has finished the parser and it returns the AST correctly. Detailed feedback is given to the students. |
| Second part | 8 | AST traversal with the Visitor design pattern [28]. |
|  | 9 | Identification phase of semantic analysis [29]. |
|  | 10 | Type checking phase of semantic analysis. |
|  | 11 | Offset computation of local and global variables, function parameters, and struct fields. |
|  | 12-14 | Code generation for a stack-based virtual machine [30]. |
|  | 15 | Code optimization. |

## Evaluation

We analyze the influence of Lex/Yacc and ANTLR generators in students' performance (students are the experimental units) with different methods. First, we measure the completion level of students' work after each lab focused on the use of lexer and parser generators (*i.e.*, labs 4 to 6 in Table 3). Such completion level is measured with a percentage indicating to what extent all the functional requirements of each lab have been met by the student. To this aim, we define a rubric and the lab instructor annotates the completion percentage for each student, after the lab. We perform the same analysis when we review their work after the first part of the course (lab 7) and for the whole compiler (lab 15).

The students have one week to finish their work before the next lab, if they do not manage to finish it in the 2 hours of each lab. At the beginning of each lab, we ask them to record the number of additional autonomous hours it took them to finish their work for the last lab. We use these data to compare how long it took the students to finish the labs related to lexing and parsing.

To gather the student's opinion about the lexer and parser generators used, we asked them to fill in the anonymous questionnaire shown in Table 4, published online with Google Forms. They filled in the questionnaire after the labs where the generators were first used (labs 4 and 5), and after implementing the first part of the language (lab 7) and the whole compiler (lab 15). Questions 4 and 5 were only asked in lab 15, since students had yet not implemented the semantic analysis and code generation modules in labs 4, 5, and 7. We compute the Cronbach's $\alpha$ coefficient [31] to measure the internal consistency of our questionnaire. The results —$\alpha = 0.862$ for labs 4, 5, and 7 and $\alpha = 0.867$ for lab 15—show its good reliability [32]. Likewise, we use the Krippendorff's $\alpha$ coefficient for ordinal data to measure inter-rater reliability [33]. Substantial reliability ($\alpha > 0.8$) is obtained for all the questionnaires but lab 15 for Lex/Yacc, which shows modest reliability ($\alpha = 0.781$) [34].

Besides measuring the students' performance and opinion after the labs, we measure their final performance. To this aim, we compare the pass and fail rates of both groups (Lex/Yacc and ANTLR) and the number of students who took the exams. We also measure and compare the marks of the lab exams and the final grades achieved by the students. We want to see if there are statistically significant differences between the values of the two groups of students. Since the two groups are independent and students' marks are normally distributed (Shapiro-Wilk test [35], p-value>0.1 for all the distributions), we apply an unpaired two-tailed *t*-test ($\alpha$

**Table 4. Questionnaire asked the students.**

| | Question |
|---|---|
| Q1 | I have found it easy to use the tool(s) to generate the lexical and syntactic analyzers. |
| Q2 | The tool(s) used to generate the lexical and syntactic analyzers are intuitive. |
| Q3 | Eventual modifications of my language specification (lexical and syntactic) will be easy to perform due to the lexer and parser generation tool(s) used. |
| Q4 | Thanks to the lexer and parser generators used, it is easier to modify the language lexer and parser than the type checker (semantic analysis). |
| Q5 | Thanks to the lexer and parser generators used, it is easier to modify the language lexer and parser than the code generation module. |

Answers are in a 5-point Likert scale, ranging from 1="completely disagree" to 5="completely agree". Questions were adapted depending on the lab delivered (e.g., for lab 4, the exact question for the Lex/Yacc group was "I have found it easy to use Lex to generate the lexical analyzer"). Thus, Q1 to Q3 only ask about lexical analysis in lab 4; for lab 5, they only ask about syntax analysis; and labs 7 and 15 include both analyses.

= 0.05) [36]—the null hypothesis ($H_0$) states that there is no significant difference between the means of the two groups.

Ever since the creation of the Programming Language Design and Implementation course, the students had implemented their lexical and syntax analyzers with the BYaccJ and JFlex generators. 2020-2021 is the first academic year we introduce the use of ANTLR. Therefore, we also analyze whether the use of a new parser generator (ANTLR) produces any significant difference with students' performance of the previous years. We conduct a statistical hypothesis test (unpaired two-tailed $t$-test) to see whether the use of ANTLR causes any difference with the Lex/Yacc approach of the previous years.

All the statistical computations were performed with IBM SPSS Statistics 27.0.1. The raw data obtained from conducting all the experiments are available for download from [37]. All the data were collected anonymously and voluntarily, and we had no access to personal data.

## Results

Table 5 shows the percentage of students who were able to complete different activities related to the implementation of lexical and syntactic analyzers. We consider the work to be completed when it fulfills at least 95% of the requirements with minor mistakes. Labs 4, 5, and 6 are focused on, respectively, lexer, parser, and AST construction. We can see how, on average, the group using ANTLR presents higher completion percentages (differences range from 5.4% to 17.3%).

Likewise, Table 6 presents the percentage of work that students managed to complete at each of the labs about lexer and parser generation. The greatest p-value obtained for the unpaired two-tailed $t$-tests is 0.015. Table 6 also presents the power of the tests. The power of a hypothesis test is the probability that the test correctly rejects the null hypothesis, denoted by

**Table 5. Percentage of students who completed different activities related to lexer and parser implementation.**

| Percentage of students who... | When | Lex/Yacc | ANTLR |
|---|---|---|---|
| Completed their work within the 2 hours of the lab | Lab 4 | 93.5% (N = 92) | 98.9% (N = 91) |
| | Lab 5 | 80.4% (N = 92) | 91.2% (N = 91) |
| | Lab 6 | 73.6% (N = 91) | 89.0% (N = 91) |
| Implemented the first part of the compiler | Lab 7 | 88.9% (N = 90) | 97.8% (N = 90) |
| Implemented the whole compiler | Lab 15 | 63.6% (N = 88) | 80.9% (N = 89) |

For each lab and group, N indicates the number of students attending the lab.

**Table 6. Percentage of work undertaken at labs 4, 5, and 6, and the number of additional hours needed to finish those labs.**

| Measure | When | Lex/Yacc | ANTLR | p-value | t-statistic | 1-$\beta$ |
|---|---|---|---|---|---|---|
| Percentage of the work completed at the lab (2 hours) | Lab 4 | 70.9% ± 2.4% | 81.9% ± 1.9% | < 0.001 | 3.545 | 0.996 |
| | Lab 5 | 79.1% ± 3.4% | 90.5% ± 3.3% | 0.012 | 2.423 | 0.934 |
| | Lab 6 | 89.8% ± 1.8% | 95.2% ± 1.1% | 0.015 | 2.472 | 0.928 |
| Number of autonomous hours it took the students to finish the lab (besides the 2 hours of each lab) | Lab 4 | 3.4 ± 0.3 | 1.7 ± 0.2 | < 0.001 | -4.845 | 0.999 |
| | Lab 5 | 6.1 ± 0.4 | 2.6 ± 0.3 | < 0.001 | -6.254 | 0.999 |
| | Lab 6 | 11.2 ± 0.6 | 8.9 ± 0.8 | 0.014 | -2.144 | 0.897 |

For each group and lab, it is shown the average values, 95% confidence intervals (standard error differences), and the p-values, t-statistics, and statistical power (1-$\beta$) of the hypothesis tests.

1-$\beta$ ($\beta$ represents the probability of making a type II error) [38]. For all the tests, we obtain values above 0.8, which is the threshold commonly taken as a standard for adequacy for the given sample sizes [38]. Therefore, we reject the null hypothesis and hence conclude that there are significant differences between the two groups. On average, ANTLR students completed from 6% to 15.5% more work in the labs than the Lex/Yacc group.

Table 6 also shows the number of additional hours it took the students to complete each lab autonomously. All the p-values for the *t*-tests are lower than 0.05 and 1-$\beta$>0.8, so differences between both groups are significant. Lex/Yacc students needed, on average, from 1.7 to 3.5 more hours than the ANTLR group.

In Table 7, we summarize the Likert scale answers for the questionnaire in Table 4. Likewise, Fig 1 shows the distribution of those answers and a visual comparison of the Lex/Yacc and ANTLR groups. The arithmetic mean of the ANTLR group is higher than Lex/Yacc, for all the questions and labs. All the ANTLR median and mode values are equal or higher than those for Lex/Yacc—92% of the median values and 64% for mode are strictly higher for ANTLR.

Fig 1 also shows how, after using different lexer and parser generators, the students think that the simplicity (Q1), intuitiveness (Q2), and maintainability (Q3) capabilities of ANTLR are greater than Lex/Yacc. For both groups, the students consider that modifying the parser is easier than modifying the semantic analyzer (Q4) and code generator (Q5), and these differences are higher for the students who used ANTLR.

The Likert scale for the questions Q1-Q5 of the questionnaire in Table 4 can be combined to provide a quantitative measure [39]. In our case, the combined values represent the student's opinion about the benefits (simplicity, intuitiveness, and maintainability) of the generator used. Fig 2 shows the sum of all the Likert values for each lab, relative to the number of questions to use the same scale (1 to 5). 95% confidence intervals are shown as whiskers. We can see that, for all the labs, there are statistically significant differences between the Lex/Yacc and ANTLR groups (*i.e.*, confidence intervals do not overlap [40] and p-values are lower than 0.01). The average values of ANTLR are from 10% to 37% higher than Lex/Yacc.

The second and third last columns in Table 8 detail some evaluation rates for the Programming Language Design and Implementation course in the 2020-2021 academic year, when we

**Table 7. Median, mode, arithmetic mean, standard deviation, and size (N) of the Likert scale answers to the questionnaire in Table 4.**

| Lab | Question | Median | | Mode | | Arithmetic mean | | Standard Deviation | | N | |
|---|---|---|---|---|---|---|---|---|---|---|---|
| | | Lex/Yacc | ANTLR | Lex/Yacc | ANTLR | Lex/Yacc | ANTLR | Lex/Yacc | ANTLR | Lex/Yacc | ANTLR |
| 4 | Q1 | 4 | 5 | 4 | 5 | 4.1 | 4.4 | 0.9 | 0.8 | 92 | 91 |
| | Q2 | 4 | 4 | 4 | 4 | 3.9 | 4.3 | 0.8 | 0.7 | | |
| | Q3 | 4 | 5 | 4 | 5 | 3.9 | 4.4 | 0.9 | 0.8 | | |
| 5 | Q1 | 3 | 4 | 4 | 4 | 3.1 | 4.4 | 1.2 | 0.5 | 91 | 90 |
| | Q2 | 3 | 4 | 3 | 4 | 3.1 | 4.3 | 1.1 | 0.6 | | |
| | Q3 | 3 | 4 | 4 | 5 | 3.3 | 4.3 | 1.1 | 0.7 | | |
| 7 | Q1 | 3 | 4 | 4 | 4 | 3.2 | 3.9 | 1.1 | 0.7 | 89 | 89 |
| | Q2 | 3 | 4 | 3 | 4 | 3.1 | 3.8 | 1.1 | 0.7 | | |
| | Q3 | 3 | 4 | 3 | 4 | 3.3 | 4.0 | 1.0 | 0.9 | | |
| 15 | Q1 | 3 | 4 | 3 | 4 | 3.4 | 4.3 | 1.0 | 0.6 | 85 | 88 |
| | Q2 | 4 | 5 | 4 | 5 | 3.3 | 4.5 | 1.0 | 0.7 | | |
| | Q3 | 3 | 5 | 3 | 5 | 3.5 | 4.6 | 0.7 | 0.5 | | |
| | Q4 | 3 | 4 | 4 | 4 | 3.3 | 4.3 | 0.9 | 0.8 | | |
| | Q5 | 3 | 4 | 4 | 4 | 3.2 | 3.7 | 1.1 | 1.1 | | |

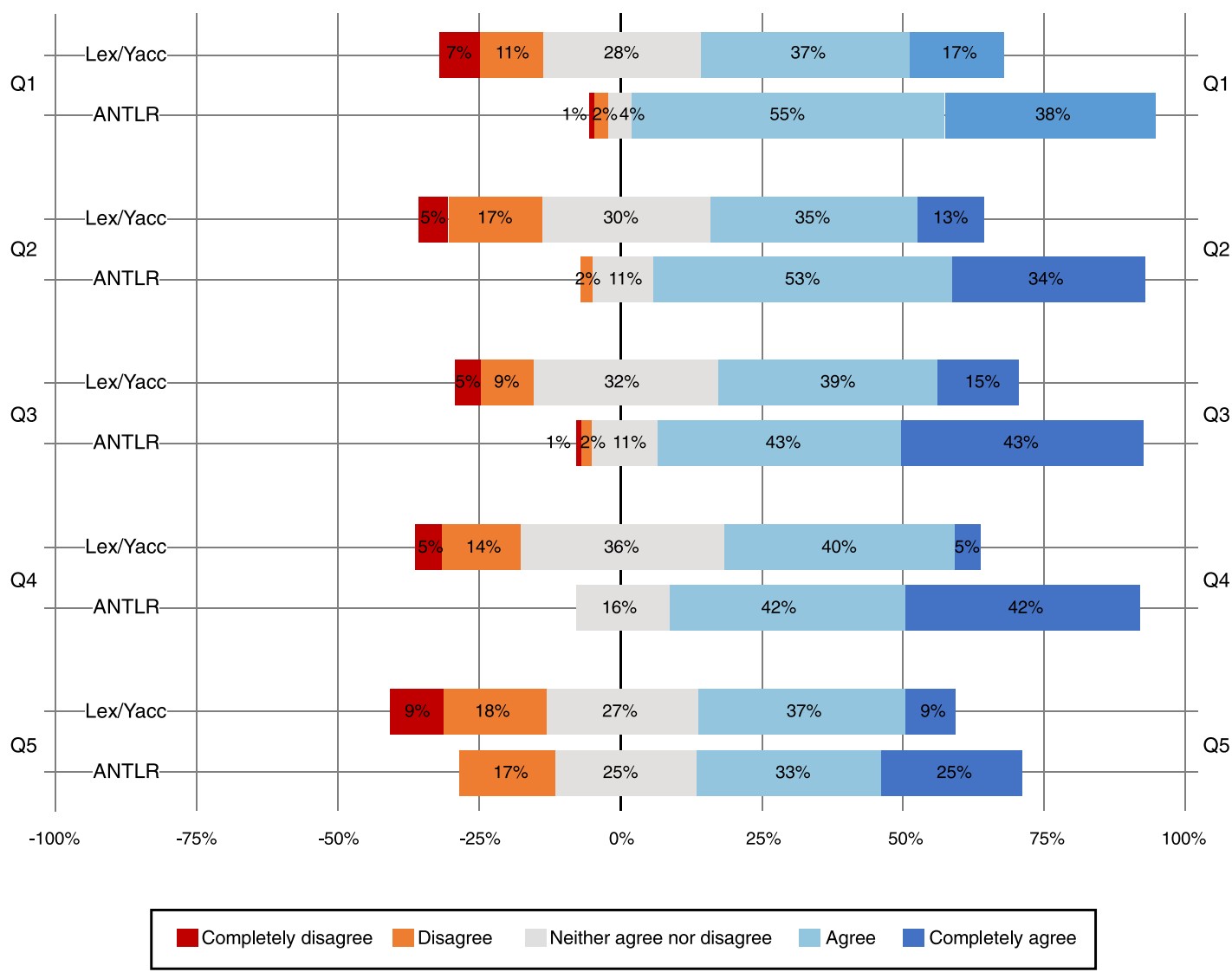

**Fig 1. Comparison of the Likert-scale answers of the Lex/Yacc and ANTLR groups.**

conducted our experiment. Those students who used ANTLR exhibit not only higher pass rates but also take more exams.

Fig 3 shows the evolution of exam attendance and pass and fail rates of the last years (for the very same course). The two last bars are the Lex/Yacc and ANTLR groups of the academic year under study. The grey area shows the 95% confidence interval for the previous seven years, to see if the values of each group are within those intervals. We can see how the Lex/Yacc group is included in the confidence interval; *i.e.*, no significant difference exists [40]. On the contrary, exam attendance and pass rates of the ANTLR group are higher than the 95% confidence interval, while the fail rate is below that interval.

Fig 4 presents the same comparison with the previous years, but considering the lab exams and the students' final marks. In the last academic year, the students of the ANTLR group obtained significantly higher marks than those in the Lex/Yacc group (95% confidence intervals do not overlap) for both the lab exams and the final marks. Compared to the previous

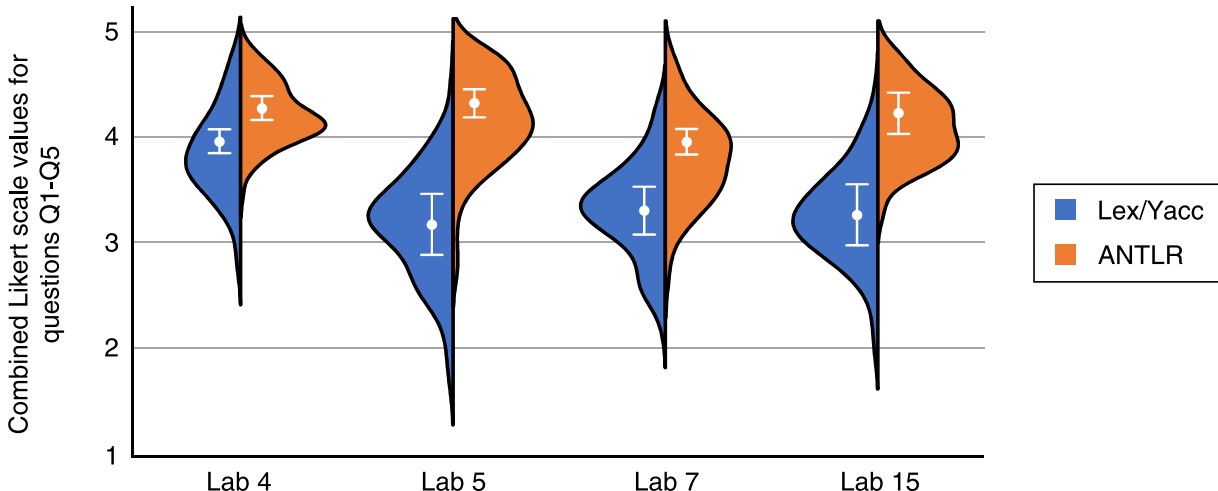

**Fig 2. Combined Likert scales for the questions Q1-Q5 of the questionnaire in Table 5.** Whiskers represent 95% confidence intervals and the points denote their mean values.

years, Lex/Yacc group shows no difference for both kinds of assessments. However, ANTLR has significantly higher values than the rest of the courses for the lab exam, and for all of them but two (2013-14 and 2016-17) in the case of the final marks. The average lab and final marks of the students of the ANTLR group are the highest ones among all the years.

## Discussion

We first discuss the students' performance when they first use the generators. For their first lexical specification (lab 4), 4.5% more ANTLR students finished their work in the lab, the percentage of completion was 15.5% higher, they require 1.7 fewer hours of autonomous work, and they consider ANTLR to be simpler, more intuitive, and maintainable than Lex. These differences are even higher after lab 5, when they build a parser. These data involve significant differences in their first contact with both lexer and parser generators.

We also consider the influence of both tools when they must be used to build a complete language parser returning an AST (lab 7) and a whole compiler (lab 15). For these two scenarios, differences are also significant. The students of the ANTLR group who managed to implement their parser and compiler on time were, respectively, 8.8% and 18.3% more than Lex/Yacc group. In their opinion, the benefits of ANTLR are higher than those using Lex/Yacc and present significant differences. This indicates that the differences detected in their first contact with the tools are maintained when they have more experience using the generators.

**Table 8. Evaluation rates in the last 8 academic years for the Programming Language Design and Implementation course.**

| Academic year | 2013-2014 | 2014-2015 | 2015-2016 | 2016-2017 | 2017-2018 | 2018-2019 | 2019-2020 | 2020-2021 | | 95% IC |
|---|---|---|---|---|---|---|---|---|---|---|
| | | | | | | | | Lex/Yacc | ANTLR | (2013–2020) |
| N (enrolled) | 42 | 61 | 101 | 114 | 142 | 162 | 162 | 92 | 91 | (68.1, 155.9) |
| Exam attendance | 70.2% | 59.0% | 67.3% | 70.2% | 72.3% | 63.0% | 64.8% | 64.8% | 81.3% | (62.2%, 70.9%) |
| Pass rate | 81.3% | 70.6% | 78.1% | 77.3% | 82.1% | 78.3% | 68.3% | 78.3% | 89,9% | (71.6%, 81.3%) |
| Fail rate | 18.7% | 29.4% | 21.9% | 22.7% | 17.9% | 21.7% | 31.7% | 21.7% | 10.1% | (18.1%, 27.8%) |
| Average mark | 7.2 | 6.4 | 6.9 | 7.4 | 6.9 | 6.8 | 6.7 | 6.6 | 7.9 | (6.5, 7.2) |

The last column shows the 95% confidence intervals of the years before our study (from 2013-2014 to 2019-2020).

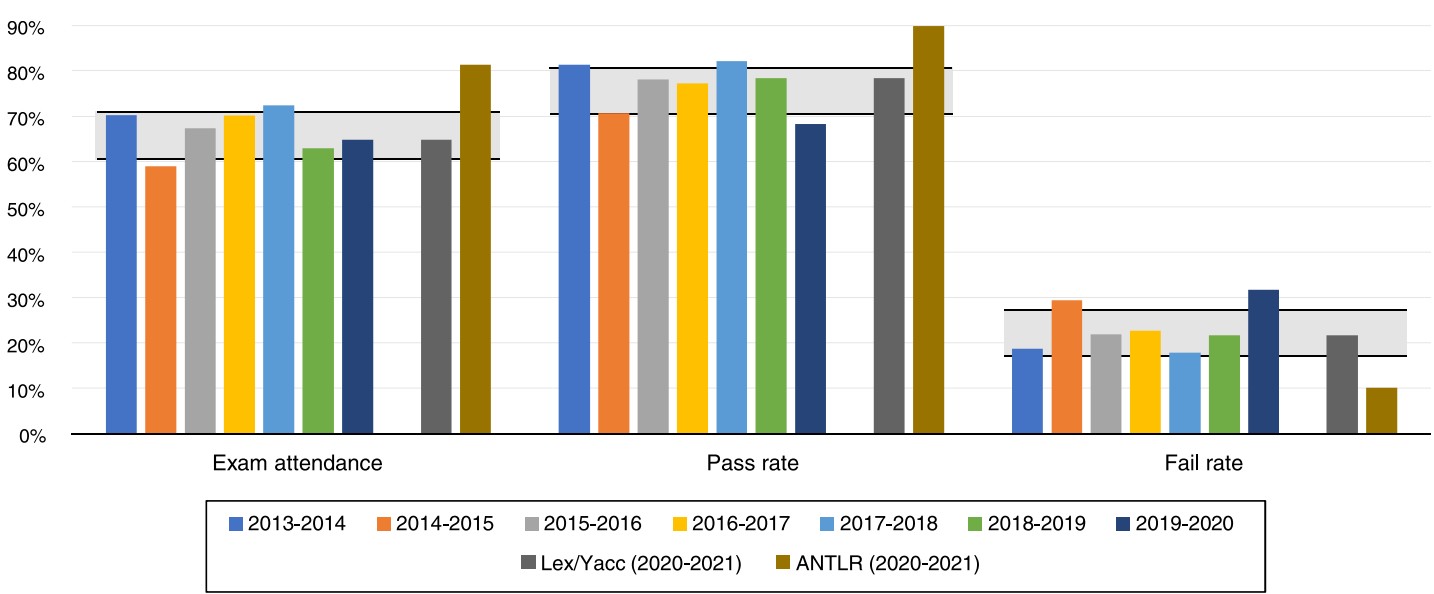

**Fig 3. Exam attendance and pass and fail rates of the last 8 academic years.** The grey area delimitates the 95% confidence intervals of the years before our study (from 2013-2014 to 2019-2020).

The use of ANTLR has also implied an increase in students' performance. For the pass rates, lab exams, and students' final marks, ANTLR show statistically significant differences, not only with the Lex/Yacc group, but also with the values of the previous academic years. This leads us to think that ANTLR eases the implementation of the programming language. It might also help to understand the theoretical concepts, but there were no significant differences in the theory marks (the higher final marks of the ANTLR group were caused by the increase in their lab marks).

The number of students taking the exam has also been increased in the ANTLR group, and it is significantly higher than in previous years. This is something important, because

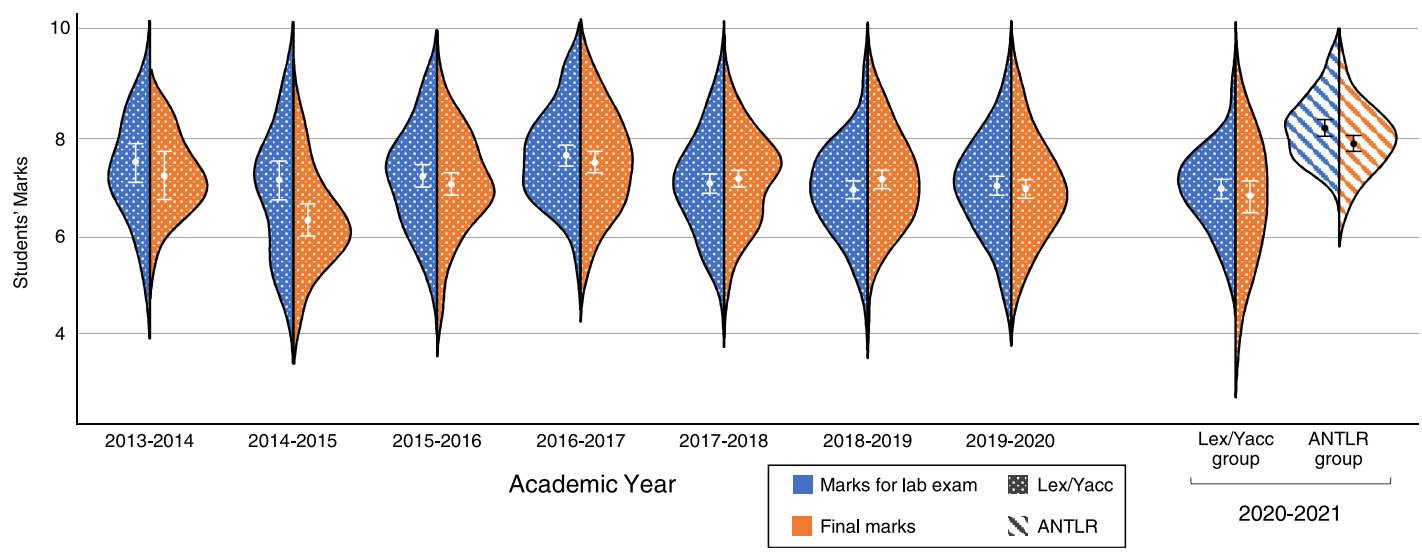

**Fig 4. Final and lab marks of the last 8 academic courses.** Whiskers represent 95% confidence intervals and the points denote their mean values.

the Programming Language Design and Implementation course has quite low rates of students taking the exams. Its average value for the past courses is 66.5%, while the ANTLR group reaches 89.9%. This fact is derived from the greater number of students finishing the language implementation (the lab exam consists in extending their language). However, it could also be interpreted as a higher confidence level to modify and extend their compiler.

Another qualitative difference detected by lecturers and students, not included in the quantitative evaluation, is the plugin support and the TestRig testing tool provided by ANTLR. With these tools, the students can interactively modify the grammar, see the parse tree for the recognized input, read the error message if the input is not correct, and see how all that changes while they modify the input. This is done within the IDE, improving the lexer/parser generation, compilation, execution, and tree visualization batch process used in the Lex/Yacc approach.

## Threats to validity

In our study, we selected two random groups of students to perform our empirical comparison. The only difference between the contents of the lectures and labs delivered to both groups was about the lexer and parser generator. Two different instructors delivered the lectures that might have influenced the differences in students' performance. However, there were no significant differences in the marks for each group in the previous years with the same instructors. For labs, most instructors teach Lex/Yacc and ANTLR groups (there are 10 different lab groups).

Regarding the differences with the previous years, one minor change is that in 2016 the weights of theory and lab marks were modified from 30%-70% to 50%-50%. In 2020, half of the course was delivered online because of the COVID-19 pandemic [41]. Nonetheless, as detailed in Table 8 and Fig 3, these changes did not cause significant changes in the students' performance. Moreover, our control group (Lex/Yacc) for the last academic year falls within the 95% confidence interval of the previous years, for all the measures (Table 8).

The conducted experiment measures the implementation of a medium-complexity imperative language. It is hard to generalize the comparison of parser generation tools for any kind of language and data format. We think the differences measured would be maintained (or even increased) for more complex languages, because ANTLR grammars tend to be more maintainable than those written in Lex/Yacc. However, it is hard to extrapolate the results to parsing scenarios of tiny languages.

Likewise, the study was conducted with Software Engineering students with Java programming skills, all of them beginners of the parser generation tools (students retaking the course were not considered). According to Zelkowitz and Wallace, this kind of validation method is a controlled experiment within a synthetic environment, where two technologies are compared in a laboratory setting [42]. Due to the validation method and the type of user used, the results should not be generalized to state that one tool is better than another one.

In an empirical research study where users utilize two different tools, the participants are aware of the tool they are using, so a blind experiment is hardly feasible [24]. Therefore, the student's expectations about the use of ANTLR may have represented a psychological factor that might have influenced their opinion of that tool. Although the lecturers did not compare ANTLR with Lex/Yacc, the students may have known they were using different tools because they could have spoken one to another. However, the measurements that are not based on student's opinion (completion percentages, work time, and evaluation data) seem to back up the answers that the students gave in the questionnaires.

## Conclusions

The empirical comparison undertaken shows that, for the implementation of a programming language of medium complexity by year-3 students of a Software Engineering degree, the ANTLR tool shows significant benefits compared to Lex/Yacc. The students show significantly higher performance when using ANTLR, for both the initial tasks of lexer and parser implementation and to modify their lexical and syntax specifications of the whole compiler. The use of ANTLR increased pass rates, exam attendance percentages, lab marks, and final grades, compared to the previous academic years when Lex/Yacc was used. According to the student's opinion, ANTLR provides higher simplicity, intuitiveness, and maintainability than Lex/Yacc.

We evaluated both tools for the development of a medium-complexity imperative language, using the Java programming language. This study could be extended to other kinds of programming languages, data formats, and implementation languages. The evaluation could also be enhanced with other parser generators [4] and different tools for compiler construction such as tree walkers [10], type checker generators [43, 44], and language implementation frameworks [45].

## Author Contributions

**Conceptualization:** Francisco Ortin.

**Data curation:** Francisco Ortin, Jose Quiroga, Oscar Rodriguez-Prieto, Miguel Garcia.

**Formal analysis:** Francisco Ortin.

**Funding acquisition:** Francisco Ortin.

**Investigation:** Francisco Ortin, Jose Quiroga, Oscar Rodriguez-Prieto, Miguel Garcia.

**Methodology:** Francisco Ortin, Jose Quiroga, Oscar Rodriguez-Prieto, Miguel Garcia.

**Project administration:** Francisco Ortin.

**Resources:** Francisco Ortin, Jose Quiroga, Oscar Rodriguez-Prieto, Miguel Garcia.

**Software:** Francisco Ortin, Jose Quiroga, Oscar Rodriguez-Prieto, Miguel Garcia.

**Supervision:** Francisco Ortin.

**Validation:** Francisco Ortin, Jose Quiroga, Oscar Rodriguez-Prieto, Miguel Garcia.

**Visualization:** Francisco Ortin.

**Writing – original draft:** Francisco Ortin.

**Writing – review & editing:** Francisco Ortin, Jose Quiroga, Oscar Rodriguez-Prieto, Miguel Garcia.

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
