## [Decision Letter · Decision Letter 0]

24 Nov 2021

PONE-D-21-34049An empirical evaluation of Lex/Yacc and ANTLR parser generation toolsPLOS ONE

Dear Dr. Ortin,

Thank you for submitting your manuscript to PLOS ONE. After careful consideration, we feel that it has merit but does not fully meet PLOS ONE’s publication criteria as it currently stands. Therefore, we invite you to submit a revised version of the manuscript that addresses the points raised during the review process.

The material in the paper looks interesting, although the manuscript should be revised carefully to meet PLOS ONE publication criteria. 

The authors should put an effort to improve the paper by taking into account the comments of all the referees, in particular those by Reviewer 5 who raised important remarks particularly related to the methodology used for the design experiments to evaluate the two tools, including the choice on the population sample used for such experiments. Please not that such reviewer was suggesting to reject the paper, however I want to give a chance to the authors to improve the manuscript by addressing the comments raised by this particular reviewer since I believe the work done is important and the raised comments are feasible to be addressed.

Please take carefully into account all the comments for improving the manuscript to meet PLOS ONE standards before resubmitting it to the journal.

We look forward to receiving your revised manuscript.

Kind regards,

Sergio Consoli

Academic Editor

PLOS ONE

Journal Requirements:

2. Please provide additional details regarding participant consent to collect personal data, including email addresses, names, or phone numbers. In the Methods section, please ensure that you have specified how consent was obtained and how the study met relevant personal data and privacy laws. If data were collected anonymously, please include this information.

"This work has been partially funded by the Spanish Department of Science, Innovation, and Universities: project RTI2018-099235-B-I00. The authors have also received funds from the University of Oviedo through its support to official research groups (GR-2011-0040)."

"This work has been partially funded by the Spanish Department of Science, Innovation, and Universities: project RTI2018-099235-B-I00. The authors have also received funds from the University of Oviedo through its support to official research groups (GR-2011-0040)."

Reviewers' comments:

Reviewer's Responses to Questions

**Comments to the Author**

1. Is the manuscript technically sound, and do the data support the conclusions?

Reviewer #1: Yes

Reviewer #2: Yes

Reviewer #3: Yes

Reviewer #4: Yes

Reviewer #5: Partly

Reviewer #6: Yes

2. Has the statistical analysis been performed appropriately and rigorously? 

Reviewer #1: Yes

Reviewer #2: Yes

Reviewer #3: Yes

Reviewer #4: I Don't Know

Reviewer #5: Yes

Reviewer #6: Yes

3. Have the authors made all data underlying the findings in their manuscript fully available?

Reviewer #1: Yes

Reviewer #2: Yes

Reviewer #3: Yes

Reviewer #4: No

Reviewer #5: No

Reviewer #6: Yes

4. Is the manuscript presented in an intelligible fashion and written in standard English?

Reviewer #1: Yes

Reviewer #2: Yes

Reviewer #3: Yes

Reviewer #4: Yes

Reviewer #5: Yes

Reviewer #6: Yes

5. Review Comments to the Author

Reviewer #1: This is a very useful paper. Empirical studies comparing the usability of different software tools are much needed and relatively uncommon. When software tools are compared, the criteria are mostly concerned with performance metrics rather than usability. This paper would be especially useful for faculty who are designing courses that make use of parser generators, but it would also be useful for software engineers in general. The paper is easy to read. There were only a few minor typos. For example, on line 48 "related related" should probably be "related work".

I did have one minor issue with the paper. The purpose of any course is for students to learn the concepts and to acquire the skill to make use of the concepts. Tools are only a means to the end, not the end itself. Unfortunately, the student performance evaluations (i.e., exam grades) combine evaluation on theory and on lab skill. It would have been useful if these two performance evaluations had been analyzed separately.

Reviewer #2: This research is of interest to researchers in language design and to instructors in software engineering and computer science. The specific question examined is fundamental to many aspects of practical programming.

Generally the work meets best practice in empirical software engineering, however there could be improvements to the way it is written up. I recommend use of the guidelines in: http://people.ucalgary.ca/~far/Lectures/SENG421/PDF/Guidelines.pdf

Reviewer #3: I'm glad to see research being done to help justify the move from established software to newer (and arguably better) alternatives. Overall, I don't find any major issues with this paper, however, a few recommendations that I'd like to see prior to its publishing:

1) I would suggest moving the related work section up towards the front of the paper and used to help motivate why ANTLR is being considered in this study.

2) In the results section, I would like to see plots of the actual distribution of the data, especially if it can be overlapped to clearly show how different each group is, and the percentage of data that lies outside the distribution of the opposing group. Furthermore, I think that there should be a better explanation of what exactly "1% work completed" means. How does that translate into non-percentage units? Are we talking 1 additional function in the assignment? 10 lines of code? 1 step of a 20 step process? Including the actual numbers alongside the percent would help to make the results a little more useful. If ANTLER allows students to get 10 more lines of code written during a lab, I might question whether or not its worth converting my entire course over to using it - but if 10 more students complete the assignment using ANTLR over Lex/Yacc, then its absolutely worth the switch.

Reviewer #4: This paper shows an empirical study on the performance of two groups of undergraduate students of software engineering implementing a compiler using Lex/Yacc and ANTLR, respectively. The authors measure students' completion rates and time spent for labs, attendance and pass/fail rates for exams, and opinions regarding the use of the tools. All data are shown in tables and figures, indicating the superiority of ANTLR over Lex/Yacc in teaching and helping students learn compiler construction more efficiently.

Overall, I think this paper is very interesting and is focused on an important issue for educators. The authors have conducted extensive data collection and quantitative analysis to support the conclusion. One interesting finding in the paper is that the choice of tools for programming assignments affects students' performance in paper exams. A shallow cause could be that students using ANTLR had more hands-on experiences, which allow them to understand the theoretical parts better. Nevertheless, a deeper one probably involves some psychological factors, which I would appreciate if the authors could include some. Another complaint is that all figures are not in place but at the end of the manuscript, causing the reading experience to be a little unpleasant.

Reviewer #5: In this manuscript, the authors conduct an empirical study to compare two widely used parser generation tools: ANTLR and Lex/Yacc. Specifically, they design experiments to evaluate the two tools from the following aspects: language implementor productivity, simplicity, intuitiveness, and maintainability. The experiment results demonstrate that ANTLR yields better performance based on the measured features.

From my point of view, the contribution of this work is twofold:

1. The authors design several metrics to measure the two parser generation tools. In particular, the metrics are defined based on the completion level of students' work, additional time that the students needed to finish the labs. Also, students' options on the tools (through questionnaire) are collected for the analysis of the tools. Those metrics (features) can help us better understand the tools.

2. The experiments are conducted in "large" scale since there are more than 90 students in each group. Also, the authors provide statistical significance evidences when comparing the differences of the two parser generation tools.

At the same time, I have two concerns about this work:

1. All the experiments are based on students from the "Programming Language Design and Implementation" course, and those students are all beginners of the tools (this is my assumption). In my opinion, we can not say one tool is better than the other tool only based on the beginners' experience.

2. When measuring the simplicity, intuitiveness, and maintainability of the tools, only one question is used to measure one aspect of the tool. For example, "Q1: I have found it easy to use the tool(s) to generate the lexical and syntactic analyzers" is used to measure the simplicity of the tool. First, I feel the question design is too general/broad, so I am not sure the answers from students are accurately measuring the specific aspect of the tool. Second, I am not sure if the students are measuring the tool in the same scale (especially when the students are not trained on how to assign the scale of the simplicity,) since the concept of simplicity (also for the other features) is different for different students. I am not convinced by the questionnaire design.

In summary, the authors design several features/metrics to compare two parser generation tools (ANTLR and Lex/Yacc) and find that ANTLR is better than Lex/Yacc based on the feedback of students from two independent groups. Even though the differences of the two tools are "significant" based on the experiment results, it is questionable whether the experiment/questionnaire is well designed (as the authors only collect the experience of beginners and the questions are too general to measure the specific aspect of a tool).

In addition one typo is found in line 48 on page 3, "the related related"?

Reviewer #6: Parsers are used various software development scenarios, two of which are Lex/Yacc and ANTLR. Surprisingly, though ANTLR provides more features, Lex/Yacc is used more in university courses settings. Different approaches exist for qualitative comparisons of the features provided by ANTLR and Lex/Yacc. However, no empirical study evaluates features such as language implementor productivity and tool simplicity, intuitiveness, and maintainability. This paper fills this gap with an empirical experiment with undergraduate students of a Software Engineering degree in a compiler design and implementation course. The students are divided into two random groups, implementing the same language with a different parser generator. The performance of both groups are statistically compared. The results show that use of ANTLR increased pass rates and exam attendance percentages.

It is amusing that ANTLR is used less than Lex/Yacc in university courses settings. This study shows the ANTLR is only used in one of fourteen top universities for compiler design and implementation course. This suggests the current status quo needs change as ANTLR has more features and is more accepted by the students in the empirical study, e.g, the students satisfaction with ANTLR, the increased passing rates and better exam attendance.

The authors comprehensively compared Lex/Yacc and ANTLR features such as parsing strategy. As with the results, the authors carefully designed a questionnaire with multiple questions for the students to evaluate the difference of the tools. This paper also contains a variety of results regarding such as the percentage of students who completed different activities related to lexer and parser implementations.

A minor point and a suggestion may be in table 1, the authors can also add top universities from USNEWS ranking.

Overall, this paper represents an original research study, in which the data supports its conclusions. It is also well-written in English. And to the best of our knowledge, it is not published elsewhere before.

6. PLOS authors have the option to publish the peer review history of their article (what does this mean?). If published, this will include your full peer review and any attached files.

Reviewer #1: **Yes: **Kenneth Baclawski

Reviewer #2: **Yes: **Mike Holcombe

Reviewer #3: No

Reviewer #4: No

Reviewer #5: No

Reviewer #6: **Yes: **Sanchuan Chen

---

## [Author Response · Author response to Decision Letter 0]

12 Jan 2022

Please, see the "response to reviewers" PDF file attached.

---

## [Decision Letter · Decision Letter 1]

9 Feb 2022

An empirical evaluation of Lex/Yacc and ANTLR parser generation tools

PONE-D-21-34049R1

Dear Dr. Ortin,

We’re pleased to inform you that your manuscript has been judged scientifically suitable for publication and will be formally accepted for publication once it meets all outstanding technical requirements.

Kind regards,

Sergio Consoli

Academic Editor

PLOS ONE

Additional Editor Comments (optional):

Reviewers' comments:

Reviewer's Responses to Questions

**Comments to the Author**

1. If the authors have adequately addressed your comments raised in a previous round of review and you feel that this manuscript is now acceptable for publication, you may indicate that here to bypass the “Comments to the Author” section, enter your conflict of interest statement in the “Confidential to Editor” section, and submit your "Accept" recommendation.

Reviewer #1: All comments have been addressed

Reviewer #2: All comments have been addressed

Reviewer #5: All comments have been addressed

Reviewer #6: All comments have been addressed

2. Is the manuscript technically sound, and do the data support the conclusions?

Reviewer #1: Yes

Reviewer #2: Yes

Reviewer #5: Yes

Reviewer #6: Yes

3. Has the statistical analysis been performed appropriately and rigorously? 

Reviewer #1: Yes

Reviewer #2: Yes

Reviewer #5: Yes

Reviewer #6: Yes

4. Have the authors made all data underlying the findings in their manuscript fully available?

Reviewer #1: Yes

Reviewer #2: Yes

Reviewer #5: Yes

Reviewer #6: Yes

5. Is the manuscript presented in an intelligible fashion and written in standard English?

Reviewer #1: Yes

Reviewer #2: Yes

Reviewer #5: Yes

Reviewer #6: Yes

6. Review Comments to the Author

Reviewer #1: (No Response)

Reviewer #2: The main issues have been dealt with. The reference to the recommended standards for empirical software engineering research are addressed.

Reviewer #5: In the updated manuscript, the authors compare two widely used parser generation tools (ANTLR and Lex/Yacc) under the academic context, and demonstrate that one tool is better than the other in the use case of students from software engineering. Compared to the previous version, the authors improved the manuscript in the following aspects:

1. The conclusion that ANTLR shows significant benefits compared to Lex/Yacc is supported by the empirical comparison results when it is in academic context. The experiments are conducted in a relatively large scale, and the statistical significance evidences are also provided. This addresses one of my concerns.

2. The authors give more information about the questionnaire design, and also utilize Krippendorff’s alpha coefficient to measure the reliability of the evaluation. As for the use of simple questions in the questionnaire, this might be the only option to collect more valid data points.

In summary, the authors did great job to improve the manuscript. Those changes address my concerns in the previous review round.

Reviewer #6: I am delighted that the authors have resolved my minor concern for last submission of this paper.

Parsers are used various software development scenarios, two of which are Lex/Yacc and ANTLR. Surprisingly, though ANTLR provides more features, Lex/Yacc is used more in university courses settings. Different approaches exist for qualitative comparisons of the features provided by ANTLR and Lex/Yacc. However, no empirical study evaluates features such as language implementor productivity and tool simplicity, intuitiveness, and maintainability. This paper fills this gap with an empirical experiment with undergraduate students of a Software Engineering degree in a compiler design and implementation course. The students are divided into two random groups, implementing the same language with a different parser generator. The performance of both groups are statistically compared. The results show that use of ANTLR increased pass rates and exam attendance percentages.

It is amusing that ANTLR is used less than Lex/Yacc in university courses settings. This study shows the ANTLR is only used in one of fourteen top universities for compiler design and implementation course. This suggests the current status quo needs change as ANTLR has more features and is more accepted by the students in the empirical study, e.g, the students satisfaction with ANTLR, the increased passing rates and better exam attendance.

The authors comprehensively compared Lex/Yacc and ANTLR features such as parsing strategy. As with the results, the authors carefully designed a questionnaire with multiple questions for the students to evaluate the difference of the tools. This paper also contains a variety of results regarding such as the percentage of students who completed different activities related to lexer and parser implementations.

Overall, this paper represents an original research study, in which the data supports its conclusions. It is also well-written in English. And to the best of our knowledge, it is not published elsewhere before.

7. PLOS authors have the option to publish the peer review history of their article (what does this mean?). If published, this will include your full peer review and any attached files.

Reviewer #1: No

Reviewer #2: **Yes: **Mike Holcombe

Reviewer #5: No

Reviewer #6: **Yes: **Sanchuan Chen

---

## [Editor Report · Acceptance letter]

14 Feb 2022

PONE-D-21-34049R1 

An empirical evaluation of Lex/Yacc and ANTLR
parser generation tools 

Dear Dr. Ortin:

I'm pleased to inform you that your manuscript has been deemed suitable for publication in PLOS ONE. Congratulations! Your manuscript is now with our production department. 

Kind regards, 

on behalf of

Dr. Sergio Consoli 

Academic Editor

PLOS ONE